# Fibroblasts Promote Resistance to KRAS Silencing in Colorectal Cancer Cells

**DOI:** 10.3390/cancers16142595

**Published:** 2024-07-20

**Authors:** Susana Mendonça Oliveira, Patrícia Dias Carvalho, André Serra-Roma, Patrícia Oliveira, Andreia Ribeiro, Joana Carvalho, Flávia Martins, Ana Luísa Machado, Maria José Oliveira, Sérgia Velho

**Affiliations:** 1i3S—Instituto de Investigação e Inovação em Saúde, Universidade do Porto, Rua Alfredo Allen 208, 4200-135 Porto, Portugal; smendonca@ipatimup.pt (S.M.O.); poliveira@ipatimup.pt (P.O.); jcarvalho@ipatimup.pt (J.C.); flaviam@ipatimup.pt (F.M.); almachado@i3s.up.pt (A.L.M.); mariajo@i3s.up.pt (M.J.O.); 2IPATIMUP—Instituto de Patologia e Imunologia Molecular, Universidade do Porto, Rua Júlio Amaral de Carvalho 45, 4200-135 Porto, Portugal; 3FMUP—Faculdade de Medicina da Universidade do Porto, Alameda Prof. Hernâni Monteiro, 4200-319 Porto, Portugal; 4ESS|P.PORTO—Escola Superior de Saúde, Instituto Politécnico do Porto, Rua Dr. António Bernardino de Almeida 400, 4200-072 Porto, Portugal; 5ICBAS—Instituto de Ciências Biomédicas Abel Salazar, Universidade do Porto, Rua Jorge de Viterbo Ferreira 228, 4050-313 Porto, Portugal; 6INEB—Instituto Nacional de Engenharia Biomédica, Universidade do Porto, Rua do Campo Alegre 823, 4150-177 Porto, Portugal

**Keywords:** KRAS, colorectal cancer, fibroblasts, cancer stemness, epithelial-mesenchymal transition, therapy resistance

## Abstract

**Simple Summary:**

Novel therapies targeting KRAS offer treatment options for previously untreatable patients. However, in colorectal cancer (CRC), resistance to KRAS-targeted therapy develops rapidly, making it imperative to understand its underlying mechanisms. Cancer-associated fibroblasts (CAFs) contribute to therapy resistance by generating and maintaining the cancer stem cell niche. This study investigates whether CAF-secreted factors induce resistance to KRAS inhibition by enhancing cancer stemness. Our findings demonstrate that while KRAS silencing reduced the expression of stem cell markers and stemness, CAF-secreted factors counteracted those effects by activating pro-tumorigenic pathways, such as epithelial-to-mesenchymal transition, and increasing cell proliferation. Overall, we provide novel mechanistic insights into how CAF-secreted factors oppose KRAS silencing-induced growth inhibition, which may be crucial for improving CRC therapy.

**Abstract:**

Colorectal cancer (CRC) responses to KRAS-targeted inhibition have been limited due to low response rates, the mechanisms of which remain unknown. Herein, we explored the cancer-associated fibroblasts (CAFs) secretome as a mediator of resistance to KRAS silencing. CRC cell lines HCT15, HCT116, and SW480 were cultured either in recommended media or in conditioned media from a normal colon fibroblast cell line (CCD-18Co) activated with rhTGF-β1 to induce a CAF-like phenotype. The expression of membrane stem cell markers was analyzed by flow cytometry. Stem cell potential was evaluated by a sphere formation assay. RNAseq was performed in KRAS-silenced HCT116 colonospheres treated with either control media or conditioned media from CAFs. Our results demonstrated that KRAS-silencing up-regulated CD24 and down-regulated CD49f and CD104 in the three cell lines, leading to a reduction in sphere-forming efficiency. However, CAF-secreted factors restored stem cell marker expression and increased stemness. RNA sequencing showed that CAF-secreted factors up-regulated genes associated with pro-tumorigenic pathways in KRAS-silenced cells, including KRAS, TGFβ, NOTCH, WNT, MYC, cell cycle progression and exit from quiescence, epithelial-mesenchymal transition, and immune regulation. Overall, our results suggest that resistance to KRAS-targeted inhibition might derive not only from cell-intrinsic causes but also from external elements, such as fibroblast-secreted factors.

## 1. Introduction

KRAS mutations occur in approximately 40% of colorectal cancers (CRCs) and are associated with a poor prognosis and resistance to therapy [1,2]. For over three decades, the KRAS oncogene has been considered an undruggable target [2,3,4], but recent breakthroughs in developing allele-specific covalent inhibitors, such as those targeting the KRAS G12C mutation, have paved the way for a new era of KRAS-targeted therapies [4,5]. These innovative therapies, now in various stages of development, show promising results by selectively inhibiting KRAS function in cancer cells [4]. Nevertheless, a disheartening reality persists: in CRC, the therapeutic response remains limited, with only a subset of patients experiencing significant benefits [2]. Furthermore, even among initial responders, resistance to treatment rapidly develops, posing a significant challenge to the long-term efficacy of these therapies [6]. While the mechanisms of acquired resistance have been extensively studied [7,8,9,10], the mechanisms of innate resistance remain poorly understood. One possible explanation is the independence of mutant KRAS cancer cells from KRAS oncogenic signaling. This has been well demonstrated in a subset of lung, pancreatic, and colorectal cancer cell lines [11,12,13,14], which remain viable upon KRAS silencing. However, clinical data shows that the most frequent outcome of KRAS-targeted inhibition in colorectal cancer is stable disease [15,16], indicating that cancer cells are indeed sensitive to KRAS inhibition to the point of stalling their growth rate. It is, therefore, imperative to better understand the mechanisms that might be involved in resistance to KRAS-targeted therapies in CRC to unveil novel combinatorial therapies that can improve their therapeutic efficacy or identify biomarkers predictive of response.

Among the several factors that have been shown to contribute to the acquisition of resistance to therapy, the tumor microenvironment, which includes cell populations such as cancer stem cells and cancer-associated fibroblasts (CAFs), is also known to drive this process [17,18,19]. Cancer stem cells possess self-renewal potential and the capacity to stay quiescent for extended periods, which confers greater resistance to various forms of therapy [20]. The development and expansion of cancer stem cells are regulated by several signaling pathways, and in CRC, this process is selectively controlled by RAS isoforms, with the KRAS isoform being the most potent inducer of stemness characteristics. KRAS activation induces stemness by up-regulating pathways such as Wnt/β-catenin and the Hedgehog, and also by increasing surface markers of stemness [21].

Even with the ongoing debate on the definitive biomarkers of stemness [22], all studies show that CRC possesses a rare cell population that resembles cancer stem cells, regardless of the markers used for isolation [23,24,25,26]. Intriguingly, these CRC cancer stem cells preferentially localize in areas enriched in CAFs [27]. This association is particularly evident in consensus molecular subtype 4 tumors (CMS4), one of the four molecular subtypes identified in CRC. These tumors are enriched in CAFs and exhibit up-regulation of genes related to cancer stem cells, highlighting their interconnection [28]. CAFs are responsible for the induction and maintenance of the cancer stem cell phenotype through their secreted factors [29,30,31,32], underscoring the co-dependence of both these cell types.

In addition to regulating the cancer stem cell phenotype, CAFs can also alter the proteome profile associated with CRC cells harboring a KRAS mutation. Specifically, we demonstrated that in colorectal cancer cells exposed to fibroblast-derived factors, KRAS oncogenic signaling is predominantly governed by fibroblast-secreted factors. However, most of the fibroblast-induced signaling is independent of mutant KRAS [33]. This suggests that CAFs might be key players in independently modulating cancer cell phenotypes in the context of KRAS-targeted therapies.

Given the symbiotic relationship between KRAS, cancer stem cells, and fibroblasts, we hypothesized that CAF-derived factors might be responsible for driving cancer cell stemness in CRC cell lines, subsequently inducing resistance to KRAS-targeted inhibition. Our results confirm that both KRAS and the secreted factors from recombinant human (rh) TGF-β1 activated fibroblasts enhance CRC stem cell activity. Furthermore, such fibroblast-derived factors recover the stemness potential lost upon KRAS silencing, leading to a more mesenchymal phenotype through the up-regulation of epithelial-to-mesenchymal transition (EMT) and pro-tumorigenic pathways. These results identify a novel potential mechanism of resistance to KRAS-target therapies, mediated by the fibroblast secretome, and open new avenues to improve the efficacy of these treatments.

## 2. Materials and Methods

### 2.1. Cell Culture

Human CRC cell lines HCT116, HCT15, SW480 (Table 1) and normal human intestinal fibroblast cell line CCD-18Co were purchased from the American Type Culture Collection (ATCC). Cells were routinely maintained at 37 °C in a humidified atmosphere with 5% CO_2_ in the recommended media: RPMI-1640 media (Gibco, Thermo Fisher Scientific, Waltham, MA, USA) for the CRC cell lines and DMEM (Gibco, Thermo Fisher Scientific, USA) for the fibroblasts, both supplemented with 10% heat-inactivated HyClone fetal bovine serum—FBS (Cytiva, Marlborough, MA, USA) and 1% penicillin-streptomycin—P/S (10,000 U/mL; Gibco, Thermo Fisher Scientific, USA).

### 2.2. Production of Conditioned Media for Fibroblasts

Fibroblasts were plated into T75 culture flasks and cultured in DMEM supplemented with 10% heat-inactivated FBS and 1% P/S at 37 °C in a humidified atmosphere with 5% CO_2_ until approximately 90% of confluence. After washing two times with phosphate-buffered saline (PBS), the media of the fibroblasts was changed to DMEM supplemented with 1% P/S plus 10 ng/mL recombinant human (rh)TGF-β1 (ImmunoTools, Friesoythe, Germany). The addition of rhTGF-β1 in the media leads to the activation of the fibroblasts, conferring on them a CAF-like phenotype. For the sphere formation experiments, the fibroblast-conditioned media was prepared with DMEM without phenol red (Gibco, Thermo Fisher Scientific, USA) in the same conditions. After 4 days in culture, the conditioned media was harvested, centrifuged at 1200 revolutions per minute (rpm) for 5 min, filtered through a 0.2 μm filter, and stored at −20 °C. The cells were harvested with 0.05% Trypsin-EDTA (Gibco, Thermo Fisher Scientific, USA), counted, and total protein extraction was performed. The confirmation of fibroblast activation was assessed through the evaluation of alpha-smooth muscle actin (α-SMA) expression by western blotting (Appendix A).

### 2.3. Cell Culture with Conditioned Media

Cancer cells were plated into a 6-well plate at a confluence of 1.5 × 10^5^ cells per well. After 16 h the cells were transfected with siRNA. Upon 6 h of transfection, the conditioned media of activated fibroblasts (grown in DMEM + 1% P/S + 10 ng/mL rhTGF-β1) was added. After 48 h of incubation with the conditioned media (total of 72 h of transfection), the cells were harvested with 0.05% Trypsin-EDTA (Gibco, Thermo Fisher Scientific, USA), counted, and collected for flow cytometry and total protein extraction. KRAS silencing efficiency was assessed by western blotting (Appendix A).

### 2.4. siRNA Transfection

Knockdown of KRAS was achieved by gene silencing using a pool of 4 small interfering RNAs (siRNA) specific for KRAS (siKRAS, ON-TARGETplus SMARTpool, from Dharmacon, GE Healthcare, Lafayette, CO, USA). This approach was previously assessed by us to determine its impact on KRAS signaling and functionality [33]. The desired cell line was plated into a 6-well plate and allowed to adhere (1.5 × 10^5^ cells per well). The following day, transfection was carried out. KRAS silencing was conducted according to manufacturer specifications using a specific ON-TARGETplus SMARTpool small interfering RNA (L-005069-00-0010; Dharmacon, GE Healthcare, USA) at a final concentration of 10 nM. As a control, a condition using non-targeting siRNA (ON-TARGETplus Non-targeting Control siRNA #1, from Dharmacon, GE Healthcare, USA) was used at the same concentration as the siRNA targeting the gene of interest. KRAS silencing efficiency was assessed by western blot after 72 h (Appendix A).

### 2.5. Protein Extraction and Western Blotting

Cells were lysed using RIPA lysis buffer [50 mM TrisHCl pH 7.5, 1% (*v*/*v*) IGEPAL CA-630, 150 mM NaCl, and 2 mM EDTA] supplemented with 1:7 protease inhibitor cocktail (Roche Diagnostics GmbH, Rotkreuz, Switzerland) and 1:100 phosphatase inhibitor cocktail (Sigma Aldrich, St. Louis, MO, USA). Cells were centrifuged at 14,000 rpm at 4 °C for 10 min. The supernatants were collected and stored at −20 °C. Protein concentration was determined using the Bradford assay (Bio-Rad Protein Assay kit, Bio-Rad, Hercules, CA, USA). Equal amounts of protein from each sample were dissolved in sample buffer [Laemmli with 5% (*v*/*v*) 2-β-mercaptoethanol and 5% (*v*/*v*) bromophenol blue] and denaturated for 5 min at 95 °C. Samples were separated in a 12% sodium dodecyl sulfate-polyacrylamide gel electrophoresis (SDS–PAGE), and proteins were transferred into nitrocellulose membranes (Amersham Protran Premium 0.45 μm nitrocellulose blotting membranes, Cytiva, USA). For immunostaining, membranes were blocked with 5% (*w*/*v*) non-fat dry milk in PBS containing 0.5% (*v*/*v*) Tween20 (0.5% PBS-T) (Sigma Aldrich, USA), and primary antibodies were incubated overnight at 4 °C with agitation (Table 2). After washing five times with 0.5% PBS-T for 5 min, membranes were incubated with HRP-conjugated anti-mouse secondary antibodies (GE Healthcare, USA) for 1 h at room temperature. Membranes were washed again five times with 0.5% PBS-T for 5 min. Bands were developed using ECL blotting substrate (Clarity Western ECL Substrate, Bio-Rad, USA). ImageJ software, version 1.54f, was used for protein quantification.

### 2.6. Flow Cytometry

For flow cytometry analysis, cells were harvested using trypsin and resuspended in RPMI supplemented with 10% heat-inactivated FBS and 1% P/S to inactivate the trypsin (complete media). Cells were allowed to recover their membrane markers through an incubation of 20 min in complete media in an incubator at 37 °C in a humidified atmosphere with 5% CO_2_. For the labeling, 2 × 10^5^ cells were used per condition. Cells were washed with wash buffer (0.5% FBS in PBS) and resuspended in that same wash buffer. Single-cell suspension was labeled using a 1:100 concentration (*v*:*v*) of a single antibody (Miltenyi Biotec, Bergisch Gladbach, Germany) in wash buffer (Table 3). As a control, a condition where no antibodies were added to the cells (unstained) was used, since it was not required to make a mix of antibodies. Fluorochrome-conjugated antibodies were incubated at room temperature, in the dark, for 15 min. Labeled cells were then rinsed in wash buffer and finally resuspended in PBS. Cells were analyzed using a FACS Canto-II (BD Biosciences) or BD Accuri C6 (BD Biosciences) flow cytometer. Data were analyzed using FlowJo version 10 cytometry analysis program. After performing a doublet exclusion gate, both the percentage of positive cells and the median fluorescence intensity (MFI) were analyzed. The gating strategy used can be found as Appendix A.

### 2.7. Sphere-Forming Assay

Following siRNA transfection, cells were harvested using Trypsin and resuspended in RPMI supplemented with 10% heat inactivated FBS and 1% P/S. Cells were centrifuged at 1200 rpm for 5 min, the supernatant was removed, and the pellet was washed with PBS. Cells were again centrifuged at similar conditions and resuspended in DMEM without phenol red and 1% P/S. Single-cell suspension was achieved by physical dissociation with a 25-gauge needle. Cells were then plated at a density of 500 cells/cm^2^ into 6-well plates coated with 1.2% poly(2-hydroxyethylmethacrylate) (Merck KGaA, Darmstadt, Germany) in 95% ethanol (Sigma-Aldrich, USA) to create non-adherent culture conditions. Cells were cultured in optimal conditions for 5 days in DMEM without phenol red containing 1× B27 and 1× N2 supplements (Life Technologies, Carlsbad, CA, USA), 20 ng/mL epidermal growth factor (EGF) (Sigma-Aldrich, USA), 10 ng/mL basic fibroblast growth factor (bFGF) (Life Technologies, USA), and 1% P/S in an incubator at 37 °C in a humidified atmosphere with 5% CO_2_. For the conditions where fibroblast-conditioned media was used, only 1× B27 and 1× N2 supplements were added because fibroblasts produce growth factors on their own [35,36,37]. Sphere-forming efficiency (SFE) was calculated as the number of spheres (≥50 μm) formed divided by the number of cells plated and multiplied by 100 to be expressed as a percentage: (SFE = Number of spheres formed/Number of cells plated × 100). To observe and acquire pictures of the spheres (in brightfield), the IN Cell Analyzer 2000 (GE Healthcare, USA) microscope was used. To automatically identify the spheres in the images, ilastik version 1.3.3, an interactive machine learning tool for (bio)image analysis as used, as well as the Cell Profiler^TM^ 4.0.7 cell image analysis software. Finally, each dataset was manually curated using the ImageJ software, version 1.54f.

### 2.8. RNA Extraction

RNA was extracted from the cells lysed with RLT+ buffer using the RNeasy Mini kit (Qiagen, Hilden, Germany) according to the manufacturer’s instructions. RNA concentration and purity were measured using the UV-Vis spectrophotometer NanoDrop 1000 (Thermo Fisher Scientific, USA). RNA was stored at −80 °C until required.

### 2.9. Library Preparation and Transcriptome Sequencing

Spheres of KRAS-silenced cells formed in the conditioned media of activated fibroblasts were compared with their KRAS-silenced counterparts, formed in control media. Due to the low sphere-forming efficiency, the quantity of RNA extracted from each experiment was low, especially for the control condition. Therefore, a pool of 3 biological replicates was used per condition. The samples were processed using Ion Torrent technology. Libraries were prepared for each sample with the Ion AmpliSeq Transcriptome Human Gene Expression Kit targeting 20,802 genes and sequenced on a 540-chip using the Ion 540 Kit-Chef and the S5 XL instrument (IonS5XL, Thermo Fisher, USA). The average mean read length was 113 base pairs. The produced reads were aligned to hg19 AmpliSeq Transcriptome ERCC v1 reference sequence and hg19_AmpliSeq_Transcriptome_21K_v1 target regions.

### 2.10. Database for Annotation, Visualization, and Integrated Discovery (DAVID)

TAC (Transcriptome Analysis Console) software was used to generate the datasets. Differential expression was analyzed with the help of DAVID online software, version 6.8 (https://david.ncifcrf.gov/home.jsp accessed on 15 July 2021). This comprehensive approach allowed a preliminary gene annotation and visualization to explore the biological functions and signaling pathways associated with both our gene sets and create tables of functional gene enrichment.

### 2.11. Gene Set Enrichment Analysis (GSEA)

Gene expression data were analyzed for enrichment using GSEA software (Broad Institute, version 4.1.0) and Human Molecular Signatures Databases (MSigDB) v2023.1.Hs, following the described guidelines [38,39,40]. Several human collections from the Molecular Signatures Database were used as the gene sets of interest, allowing us to explore the enrichment of biologically relevant pathways and functional annotations. The normalized enrichment score (NES) for each gene was calculated and used for further analysis and graphical representation.

### 2.12. Quantification of G0 Arrest

The quantification of the G0 arrest in the RNAseq data was adapted from the G0 arrest score quantification described in [41]. R software was used for gene enrichment and determination of G0 arrest score (original code available at: https://github.com/secrierlab/CancerG0Arrest/tree/main/TCGA_QuiescenceEvaluation accessed on 15 October 2023). A final positive G0 arrest score indicates that cells are quiescent, while if negative, cells are cycling.

### 2.13. Statistical Analysis

Results are representative of three or more independent experiments. Quantifications are expressed as mean ± standard deviation (SD) of the biological replicates considered. Statistical analyses were performed using GraphPad Prism v8 (GraphPad Software Inc., USA). All samples were tested for normality, and significant statistical difference was considered when the *p* value was less than 0.05. The statistical tests performed for each analysis are indicated in the corresponding figure legend.

## 3. Results

### 3.1. Colorectal Cancer Cells Express Variable Basal Levels of Membrane Cancer Stem Cell Markers

To explore the role of mutant KRAS in modulating the cancer stem cell landscape in CRC, we selected three CRC cell lines (HCT116, HCT15, and SW480), all harboring a KRAS mutation (Table 1), though with different origins and genetic profiles. We began by characterizing the basal stemness potential of each cell line by flow cytometry analysis, focusing on the most commonly used membrane stem cell markers: CD24, CD133, CD166, and CD44 and its isoform CD44v6 (Table 3). Additionally, we also included CD49f (Integrin α6) and its binding partner CD104 (Integrin β4) in our analysis. CD49f, in particular, has been described as a biomarker transversally present amongst stem cell populations, including the intestinal one [42], an enhancer of tumorigenesis [23], and a possible regulator of other stem cell markers, such as CD44 [43].

Our analysis revealed a heterogeneous expression of cancer stem cell markers within and across the CRC cell lines, both in the percentage of positive cells and in the level of expression per cell, denoted by the median fluorescence intensity (MFI) (Figure 1). Interestingly, only CD49f and CD104 receptors were highly and consistently expressed across the three CRC cell lines.

### 3.2. KRAS Silencing Up-Regulates CD24 and Down-Regulates CD49f and CD104 Stemness Markers across Cell Lines

Having established the basal expression levels of the stemness markers, we next investigated the role of KRAS in regulating this stemness signature. To do so, we silenced the expression of KRAS by RNAi in the three cell lines, followed by flow cytometry analysis of the abovementioned stem cell markers.

Silencing KRAS led to significant alterations in the expression of stem cell markers, predominantly reflected in changes in the MFI rather than the percentage of positive cells (Figure 2). In the HCT15 cells, KRAS silencing up-regulated CD24 and CD44 MFI, while decreasing the MFI of CD49f and CD104. CD44v6, CD133, and CD166 remained unchanged. In the HCT116 cells, KRAS silencing increased the percentage of CD24-positive cells. On the other hand, CD49f MFI decreased, along with both the MFI and percentage of positive cells, for CD104, CD44, CD44v6, and CD133. CD166 remained unchanged. In the SW480 cells, KRAS silencing up-regulated CD24 and the MFI of CD133 and CD166, whereas the MFI of CD49f, CD104, and CD44 decreased. No alterations were found in CD44v6 expression.

In summary, KRAS silencing induced significant, cell line-dependent changes in the expression of the stem cell markers. Despite this apparent cell line dependence, consistent alterations in CD24, CD49f, and CD104 expression were evident across all cell lines: after KRAS silencing, there was an up-regulation of CD24 (except the MFI in HCT116) while the MFI of CD49f and CD104 decreased. These changes strongly suggest a decrease in cancer stemness potential upon KRAS silencing.

### 3.3. Fibroblast-Secreted Factors Attenuate the Capacity of KRAS Silencing to Regulate the Expression of Cancer Stem Cell Markers

We then examined whether the factors secreted by activated fibroblasts could influence the modulatory effects of KRAS on stem cell markers. KRAS-silenced cells were cultured with conditioned media from activated fibroblasts (CCD-18Co normal colon cell line treated with rhTGF-β1). We then analyzed the abovementioned stem cell markers by flow cytometry.

Strikingly, treatment with conditioned media of activated fibroblasts attenuated the differences in cancer stem cell expression previously observed between the control and KRAS-silenced cells (Figure 3). After KRAS silencing and treatment with conditioned media in the HCT15 cells, CD24 expression increased, as seen with KRAS silencing alone, but the reduction in the MFI of CD49f and CD104 was no longer evident. Furthermore, CD44 expression decreased, contrary to KRAS silencing alone. CD44v6, CD133, and CD166 remained unchanged. In HCT116 cells, all significant differences between control and KRAS-silenced cells were lost after treatment with conditioned media. Similarly, in SW480 cells, conditioned media annulled the differences in most markers, though CD44 reduction remained. CD44v6 expression decreased, contrary to KRAS silencing alone.

These findings highlight the role of fibroblast-secreted factors in shaping the response of CRC cells to KRAS silencing. They suggest that fibroblast-secreted factors can mitigate the effects of KRAS silencing on stem cell marker levels in the membrane of CRC cells, thus unveiling another layer of intricacy in the interplay between KRAS signaling and fibroblasts.

### 3.4. Fibroblast-Secreted Factors Attenuate the Inhibitory Effect Promoted by KRAS Silencing in the Sphere Formation Assay

To further understand how KRAS modulation and fibroblast-derived signals influence the stem cell-like phenotype, we performed an in vitro sphere formation assay, a technique commonly used for assessing stem cell potential based on the capacity of cells for self-renewal. For this assay, cells were plated in anchorage-free conditions. Since resistance to anoikis is a characteristic of stem cells, the higher the number of spheres formed, the greater the stem cell potential of that cell line [44].

We started by investigating the role of KRAS silencing alone. KRAS silencing reduced sphere-forming efficiency (SFE) across all cell lines, consistent with the decreased stem cell marker expression observed earlier by flow cytometry (Figure 4a).

Subsequently, we tested the effect of fibroblast-secreted factors on the SFE of control and KRAS-silenced cells. Treatment with fibroblast-conditioned media enhanced sphere formation in all cell lines, including control and KRAS-silenced cells. Except for SW480, conditioned media of fibroblasts abolished the decrease in SFE induced by KRAS silencing (Figure 4b).

These results suggest that fibroblast-secreted factors not only enhance the self-renewal and proliferation capacity of CRC cell lines but also confer resistance to KRAS silencing.

### 3.5. Fibroblast-Secreted Factors Up-Regulate Proliferation, Pro-Tumorigenic, EMT, and Immune System Regulation Pathways in KRAS-Silenced Cells

To uncover the molecular mechanisms by which fibroblast-conditioned media counteract KRAS silencing, we conducted an RNASeq analysis. We focused on the HCT116 cell line since it showed the highest SFE increase in the KRAS-silenced condition after treatment with conditioned media of fibroblasts. We compared the gene expression profiles of KRAS-silenced cells treated with conditioned media versus control media to identify specific genes and pathways influenced by fibroblast-derived factors.

Analysis of the hallmark gene sets in the human molecular signature database revealed that KRAS-silenced cells treated with conditioned media exhibited a distinctive gene expression profile (Figure 5a–c and Appendix A). Treatment with conditioned media of activated fibroblasts up-regulated pathways associated with cell cycle control (E2F targets, G2-M checkpoints, MYC targets, mitotic spindle pathways), EMT, immune system regulation (IL6, IL2, interferon-gamma, complement pathways), and several pro-tumorigenic pathways (KRAS, NOTCH, TGF-β, and WNT).

In contrast, treatment with control media up-regulated pathways related to metabolism (cholesterol, fatty acids, and xenobiotic metabolism), cellular stress (apoptosis and hypoxia, TNF-α and mTOR signaling pathways), and DNA damage response (unfolded protein response and P53 pathways).

Since we had up-regulation of pathways related to cell cycle control after treatment with conditioned media of fibroblasts, we evaluated the G0 arrest transcriptional signature. This methodology distinguishes between cells in a state of G0 arrest (encompassing quiescence, senescence, and dormancy) and those in a rapid cell cycle progression state [41]. This analysis revealed that fibroblast-conditioned media drove KRAS-silenced cells into a state of rapid cell cycling (negative G0 score), akin to stem cell self-renewal, while control media maintained cells in a quiescent state (positive G0 score) (Figure 5d).

These results indicate that fibroblast-secreted factors override the growth-inhibitory effects of KRAS silencing, enhancing proliferative potential and promoting a mesenchymal phenotype through EMT and stemness-related pathways.

## 4. Discussion

In an era of intensive research to develop effective therapeutic strategies against KRAS mutant tumors, our in vitro study uncovers a crucial mechanism that bypasses KRAS silencing. This mechanism operates independently of KRAS and is orchestrated by CAF-derived factors that trigger cancer stem cell activity, proliferation, EMT, modulation of immune response, and up-regulation of tumorigenic pathways.

KRAS, located downstream of many cell surface receptors, is a central regulator of intracellular signaling in response to extracellular stimuli, making it crucial for regulating diverse cancer cell activities, including stemness [2,33,45]. This role of KRAS could be exploited, since targeting cancer stem cells to prevent cancer recurrence remains one of the most significant challenges in oncology. Therefore, understanding the factors and circumstances that regulate cancer stem cell induction and maintenance—and consequently govern the development of recurrence—is essential for improving treatment efficacy [18].

We found that while the expression of stem cell markers varied in a cell line-dependent manner, KRAS silencing consistently resulted in up-regulation of CD24 and down-regulation of CD49f and CD104 across cell lines. Although CD24 up-regulation has been previously correlated with tumor progression, invasiveness, differentiation, and chemotherapy resistance [46,47,48], its role in modulating the CRC stem cell phenotype remains inconclusive. Some studies suggest that loss of CD24 expression is associated with poorer outcomes [49], while others suggest that CD24 is a good prognosis marker in CRC, being down-regulated in stage IV colorectal adenocarcinoma [46]. Regarding the expression of CD49f/CD104, our findings align with existing literature: both CD49f and CD104 were highly expressed at basal levels across the CRC cell lines analyzed. The CD49f/CD104 complex is crucial for cell-cell and cell-matrix interactions and is typically overexpressed in CRC. Depletion of the integrin CD49f/CD104 complex reduces the invasive and migratory capabilities of cancer cells [50], highlighting its role in promoting an aggressive phenotype. Although the mechanisms controlling its expression are still not fully known, transcription factors, such as MYC, have been implicated in this process. Our data reinforces KRAS as a regulator of CD49f/CD104 expression in CRC cells. KRAS silencing led to a reduction in the expression of these markers, indicating a decreased stemness potential, as confirmed by our functional assay that showed a reduction in SFE. However, treatment with conditioned media from fibroblasts annulled the effects of KRAS silencing. It restored the expression of CD49f and CD104, indicating an increase in stemness potential, also confirmed by the enhanced SFE. Accordingly, CD49f/CD104 can increase cell proliferation through the activation of the Wnt/β-catenin pathway, a regulator of stem cell homeostasis in the intestinal crypts and an upstream effector of MYC [51].

Our RNAseq results further reinforced the role of fibroblasts as mediators of resistance to KRAS silencing. Fibroblast-derived factors induced the up-regulation of multiple pathways linked to critical cellular processes such as cell cycle control, EMT, immune system regulation, and tumor development. Among these pathways, there were key signaling routes such as KRAS, NOTCH, TGF-β, MYC, and WNT known to play significant roles in cancer progression and therapy resistance. Specifically, E2F, MYC, and G2M pathways, often down-regulated upon KRAS inhibition [52,53], were among the top-ranked processes in KRAS-silenced cells stimulated by fibroblast-secreted factors. Up-regulation of these pathways in the presence of fibroblast-secreted factors suggests a mechanism by which fibroblasts can counteract the effects of KRAS silencing, promoting cell cycle progression and proliferation despite the absence of KRAS activity. This was further validated by a G0 arrest transcriptional signature analysis, revealing that KRAS-silenced cells treated with fibroblast-conditioned media shift from a quiescent state to active cell cycling, similar to stem cell renewal.

Interestingly, the stem cell-like phenotype and EMT are deeply intertwined, as cells undergoing EMT can acquire cancer stem cell properties, while cancer stem cells themselves can undergo EMT to facilitate metastasis [54]. One way of inducing EMT is through the activation of the canonical TGF-β pathway. TGF-β signaling has been correlated with CRC subtypes with a worse prognosis and increased relapse. Conditioned media of CAFs are enriched in TGF-β1 and, therefore, can induce EMT [55]. It also contains other factors such as fibroblast growth factor (FGF), interleukin-6 (IL-6), hepatocyte growth factor (HGF), osteopontin (OPN), and stromal-derived factor-1α (SDF1) [54]. The latter three can modulate cancer cells into a more stem cell-like phenotype, particularly through the Wnt/β-catenin pathway, increasing CD44v6 expression. IL-6 plays a critical role in immune regulation by promoting a chronic inflammatory environment and controlling NOTCH activation, a pathway responsible for intestinal stem cell self-renewal, cancer stem cell maintenance, TGF-induced EMT, and therapy resistance [54,56,57]. In our study, NOTCH3 was one of the top up-regulated genes in the NOTCH pathway, commonly aberrantly expressed in human cancers. In CRC, NOTCH3 expression increases with tumor staging and is correlated with worse prognosis, poor overall survival, and CMS4 tumors, which are CAF-enriched and also associated with TGF-β signaling [58,59]. NOTCH3 also plays a role in supporting cancer stemness and resistance to therapy [59], which can be circumvented by the use of NOTCH inhibitors since they reduce the expression of stem cell markers and improve response to chemotherapy and radiotherapy [57]. Therefore, our data opens the door for future studies addressing the effects of combining NOTCH inhibitors, such as gamma-secretase inhibitors, with KRAS-targeting therapies as a strategy to mitigate CAF-induced stemness and resistance to KRAS inhibitors.

Another way of inducing EMT by the canonical TGF-β pathway involves the activation of the Hippo pathway [60] through the regulation of several of its key effectors. One crucial interaction is between SMAD proteins, activated by TGF-β signaling, and YAP/TAZ, the primary effectors of the Hippo pathway. These proteins translocate to the nucleus, where they work together to regulate gene expression, controlling transcription of genes involved in EMT, cell proliferation, and survival [61]. KRAS-targeted inhibition (using inhibitors such as sotorasib or adagrasib) can also inadvertently activate Hippo signaling [62], which promotes the maintenance of EMT characteristics essential for cancer cells to overcome KRAS suppression and develop resistance against these inhibitors [60,62]. Our RNAseq results on KRAS-silenced cells align with the literature. Cells treated with conditioned media of fibroblasts show an up-regulation of genes in the Hippo pathway, including YAP1, WWTR1/TAZ, and TEAD1, when compared with control treatment (Appendix A). It is, therefore, plausible that in cancers harboring KRAS mutations, CAF-secreted factors can enhance the Hippo signaling pathway through the mediation of the TGF-β pathway. This interplay could further enhance EMT, facilitating tumor progression and resistance to KRAS-targeted therapies. More studies are required to further explore this putative mechanism and identify possible options to disrupt these pathways, improving therapeutic outcomes.

CAFs also play a significant role in regulating cancer metabolism, significantly impacting the metabolic landscape of the tumor microenvironment [63]. Alongside this, oncogenic KRAS also changes tumor metabolism to support quick proliferation and survival by increasing glucose uptake, enhancing glycolysis, and reprogramming lipid metabolism [64,65]. Interestingly, our RNAseq analysis revealed a down-regulation of pathways related to cholesterol, fatty acids, and xenobiotic metabolism in KRAS-silenced cells treated with fibroblast-conditioned media. Our results indicate that fibroblasts might be aiding the metabolic needs of cancer cells by providing them with substrates and signaling molecules, which can help them overcome the metabolic stress caused by KRAS inhibition. It is possible that cancer cells take up these metabolites to power oxidative phosphorylation and other growth processes. This cooperation allows cancer cells to reduce their own cholesterol and fatty acid production, relying instead on the metabolites provided by the CAFs [63]. The reduction in xenobiotic metabolism may also indicate that these cancer cells are conserving energy, focusing on growth and proliferation while leaving detoxification to the CAFs. Further studies are required to dissect the mechanisms behind this metabolic shift and identify its molecular mediators, guiding the exploration of combination therapies that target both metabolic adaptations and KRAS.

Our data pinpoints fibroblast infiltration as a potential biomarker for predicting a lack of response to KRAS-targeted treatments and lifts the veil on a fibroblast-mediated mechanism of resistance to KRAS inhibitors (Figure 6). Such insights can be particularly important in the case of CMS4 CRC, which is highly infiltrated by fibroblasts and might inherently resist KRAS inhibition. Furthermore, we previously showed that KRAS-silenced HCT116 cells promote fibroblast migration and activation [45]. When put together with our current findings, this suggests a feedback loop driving acquired resistance: fibroblast-poor CRCs, upon KRAS inhibition, recruit and activate fibroblasts into CAFs, which in turn support CRC growth independently of KRAS signaling. Our research underscores the critical need to explore these mechanisms further to improve the clinical management of KRAS mutant CRC.

## 5. Conclusions

In conclusion, our findings underscore the need to rethink current therapeutic strategies for targeting KRAS-mutant CRCs. Targeting KRAS alone appears insufficient to tackle tumor growth effectively. CRC patients could benefit significantly from combined therapies that target not only KRAS but also the cancer-associated fibroblasts (CAFs) within the tumor microenvironment. Furthermore, our results highlight several potential CAF-derived molecular targets that merit further exploration as therapeutic interventions. By adopting a more comprehensive approach that addresses both cancer cells and their supportive environment, we can pave the way for more successful treatments for KRAS-mutant CRC.

## Figures and Tables

**Figure 1 cancers-16-02595-f001:**
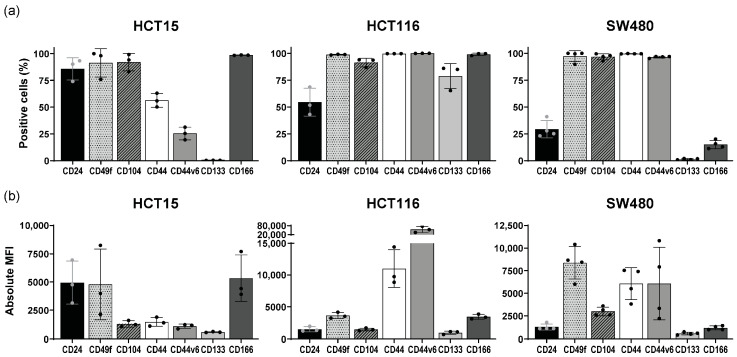
Characterization of the basal levels of stem cell marker expression in CRC cell lines by flow cytometry. Mean and standard deviation are represented in each bar. Each dot represents a biological replicate. (**a**) Percentage of positive cells; (**b**) Absolute median fluorescence intensity (MFI).

**Figure 2 cancers-16-02595-f002:**
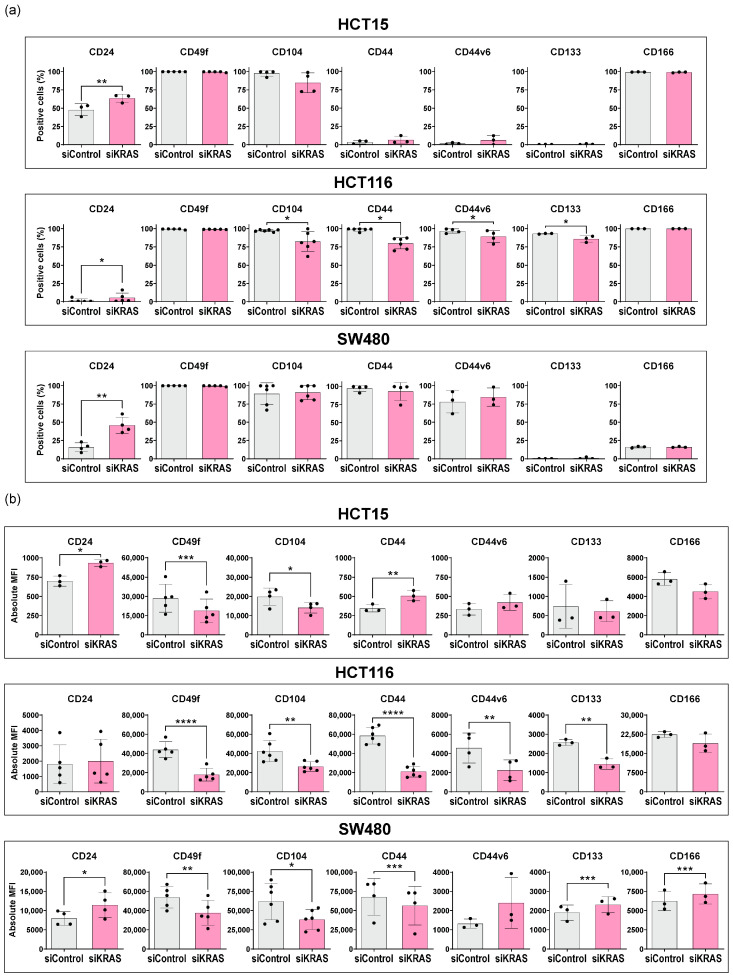
Stem cell marker expression by flow cytometry in CRC cell lines after KRAS silencing. For all cell lines, normality of the data was tested using Shapiro-Wilk normality test. A one-tailed paired *t*-test was performed, testing for a *p*-value < 0.05. The symbols *, **, ***, and **** were used to denote levels 0.05, 0.01, 0.001, and 0.0001 of statistical significance, respectively. For the samples that did not follow normality, a Wilcoxon matched-pairs signed rank test was used. Mean and standard deviation are represented in each bar. Each dot represents a biological replicate. (**a**) Percentage of positive cells; (**b**) Absolute median fluorescence intensity (MFI).

**Figure 3 cancers-16-02595-f003:**
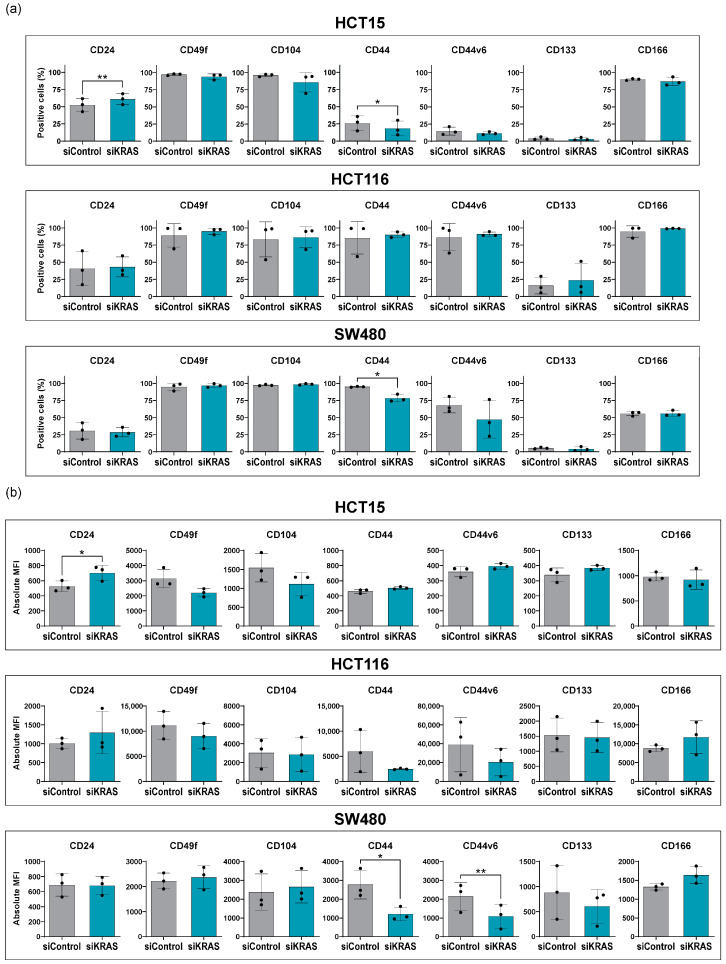
Stem cell marker expression by flow cytometry in CRC cell lines after KRAS silencing, plus treatment with conditioned media from fibroblasts. For all cell lines, normality of the data was tested using Shapiro-Wilk normality test. A one-tailed paired *t*-test was performed, testing for a *p*-value < 0.05. The symbols * and ** were used to denote levels 0.05 and 0.01 of statistical significance, respectively. For the samples that did not follow normality, a Wilcoxon matched-pairs signed rank test was used. Mean and standard deviation are represented in each bar. Each dot represents a biological replicate. (**a**) Percentage of positive cells; (**b**) Absolute median fluorescence intensity.

**Figure 4 cancers-16-02595-f004:**
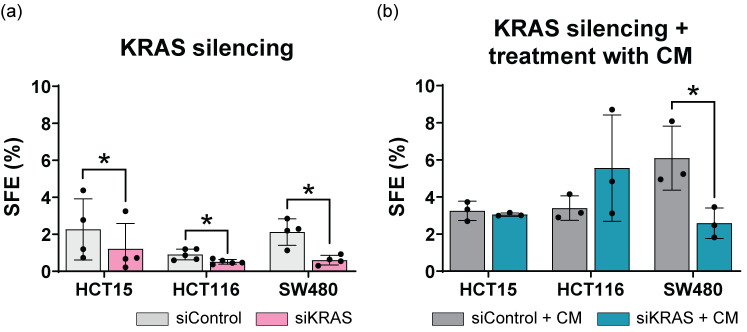
Sphere formation assay of CRC cell lines after silencing with siControl or siKRAS. For all cell lines, normality of the data was tested using a Shapiro-Wilk normality test, and a one-tailed Paired *t*-test was performed, testing for a *p*-value < 0.05 (symbol * denotes level 0.05 of statistical significance). Mean and standard deviation are represented in each bar. Each dot represents a biological replicate. (**a**) Sphere forming efficiency (SFE) percentage after treatment with sphere formation assay medium; (**b**) SFE percentage after treatment with conditioned media from activated fibroblasts (CM).

**Figure 5 cancers-16-02595-f005:**
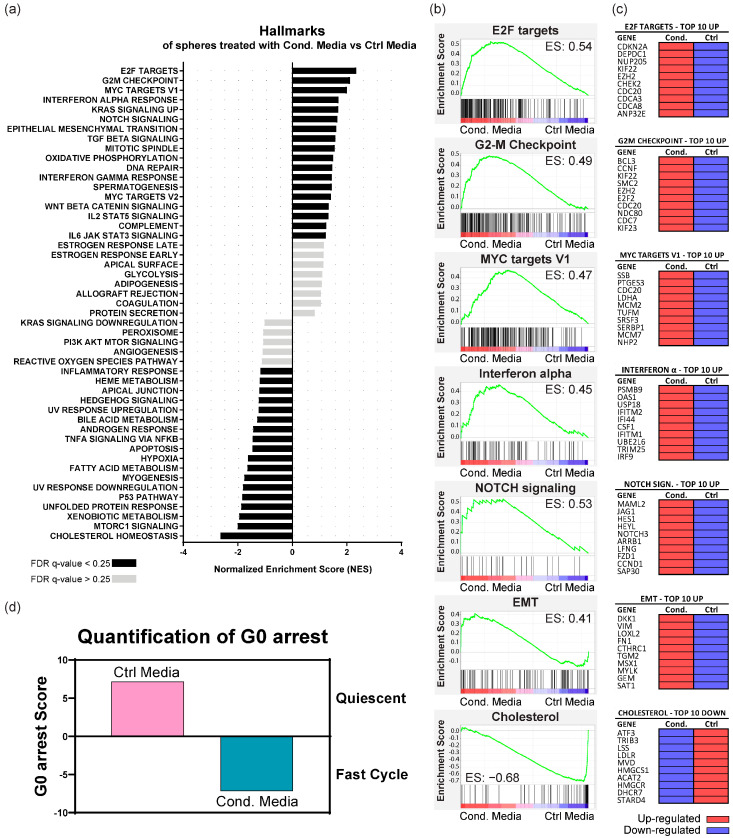
RNAseq analysis of KRAS-silenced HCT116 spheres treated with conditioned media of activated fibroblasts (Cond. Media) against control media (Ctrl Media). (**a**) Results of GSEA Hallmark analysis showing enriched gene sets. Black bars indicate significant enrichment at a false discovery rate (FDR) < 25%, while gray bars represent gene sets with FDR > 25%. A positive normalized enrichment score (NES) value indicates enrichment in the cells treated with conditioned media of fibroblasts, whereas a negative NES indicates enrichment in the cells treated with control media; (**b**) Enrichment plots for the most relevant data sets enriched in GSEA Hallmark analysis, showing the profile of the running enrichment score (ES) and positions of gene set members on the rank-ordered list; (**c**) Tables showing the top 10 enriched genes in each data set. Red indicates up-regulation of the gene, while purple indicates down-regulation; (**d**) Quantification of G0 arrest. Positive values indicate that cells are in a quiescent state, whereas negative values identify cells that are in a proliferative cycle phase.

**Figure 6 cancers-16-02595-f006:**
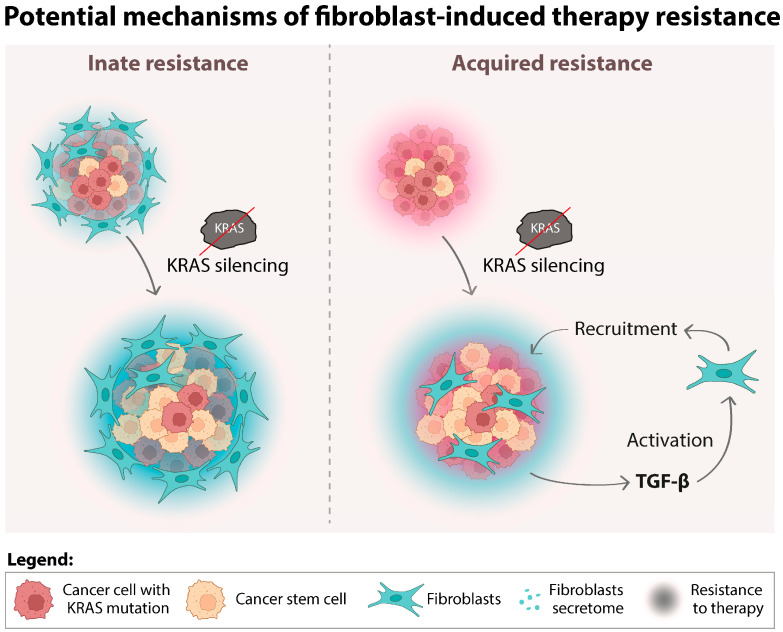
Fibroblast-mediated resistance to KRAS-targeted therapies: possible clinical implications. The combination of our current and previous results leads us to hypothesize two possible forms of resistance mechanisms mediated by the action of fibroblasts: innate and acquired. Innate resistance occurs in fibroblast-rich tumors, such as those from CMS4, that can inherently withstand KRAS inhibition through the support provided by resident fibroblasts. Conversely, acquired resistance may develop in initially fibroblast-poor tumors that recruit and activate fibroblasts in response to KRAS inhibition, altering the tumor microenvironment to its advantage. Both scenarios ultimately lead to resistance to KRAS-targeted therapies.

**Table 1 cancers-16-02595-t001:** Genetic and histological characterization of KRAS mutant CRC cell lines. Adapted from [34].

Cell Line	KRAS Mutation	Disease	Derived from	Reference
HCT116	G13D	Colorectal Carcinoma	Primary tumor	[34]
HCT15	G13D	Colorectal Adenocarcinoma	DLD-1 misclassified	
SW480	G12V	Colorectal Adenocarcinoma	Primary tumor	

**Table 2 cancers-16-02595-t002:** List of antibodies used for western blotting.

Ab ^1^	MW ^2^ (kDA) ^3^	Blocking	Dilution	Species	Brand	Catalog no.	Storage (°C)	Secondary Dilution
α-SMA	42	5% milk in 0.5% PBS-T	1:250	Mouse	Abcam	ab7817	−20	1:3000
GAPDH	37	5% milk in 0.5% PBS-T	1:10,000	Mouse	SantaCruz	sc-47724	4	1:20,000
KRAS	21	5% milk in 0.5% PBS-T	1:4000	Mouse	LSBio	C175665	−20	1:8000

^1^ Antibody. ^2^ Molecular weight. ^3^ Kilodalton.

**Table 3 cancers-16-02595-t003:** List of anti-human antibodies used for flow cytometry.

Antibody	Conjugate	Clone	Catalog no.	Brand
CD24	PE	32D12	130-098-861	Miltenyi Biotec
CD49f	APC	GoH3	130-100-147	Miltenyi Biotec
CD104	FITC	REA236	130-124-266	Miltenyi Biotec
CD44	FITC	DB105	130-113-896	Miltenyi Biotec
CD44.V6	APC	REA706	130-111-425	Miltenyi Biotec
CD133	PE	AC133	130-098-826	Miltenyi Biotec
CD166	APC	REA442	130-106-619	Miltenyi Biotec

## Data Availability

The original data presented in the study are openly available in https://www.mdpi.com/article/10.3390/cancers16142595/s1, accessed on 29 May 2024 or article/Appendix A. Further inquiries can be directed to the corresponding author.

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
