# Peer review of "Fibroblasts Promote Resistance to KRAS Silencing in Colorectal Cancer Cells"

_cancers, 2024, doi:10.3390/cancers16142595_

Round 1

Reviewer 1 Report

Comments and Suggestions for Authors

An article by Dr. Velho and the group elaborating on the Mechanisms to Bypass KRAS Inhibition and focusing on the Influence of Fibroblast-Secretome on this aspect makes this article a novel experimental approach with translational application for future therapies. Though a few things must be addressed before it is ready for acceptance, they are as follows:

1. It has been known that KRAS and Hippo signaling converge to regulate EMT and tumor survival (PMID: 24954536), and it has also been shown recently that KRAS inhibition with small molecule inhibitors (sotorasib or adagrasib) activates Hippo signaling, which in turn creates resistance against the inhibitors (PMID: 37729426). Since this article, it has been shown that CAF-secreted factors enhanced EMT signaling and promoted proliferation. This clearly suggests CAF might play a role in enhancing Hippo signaling, which also activates EMT in KRAS-driven cancers.  This aspect should be addressed as one of the future topics of this study to be followed up on. Add a few lines to the discussion on this topic by referring to the relevant references above. 

2. CAF has also been known to regulate cancer metabolism (PMID: 33540679), and oncogenic KRAS mediates altered tumor metabolism. This aspect is another topic that might be followed up on in future studies generated from the current article. Authors should add a few lines on this by adding a few lines and the relevant mentioned reference.

3. Figure 5, use different colors in the graphs to make the prominent distinctions between variables. 

Author Response

Overall: We appreciate the reviewer's positive feedback regarding the quality of our written English and the overall quality of our work. In response to the comment about enhancing the introduction with more background and relevant references, we have thoroughly revised the introduction chapter. Not only we added more content and relevant references, but input a more narrative flow into the chapter.

- Comments 1: It has been known that KRAS and Hippo signaling converge to regulate EMT and tumor survival (PMID: 24954536), and it has also been shown recently that KRAS inhibition with small molecule inhibitors (sotorasib or adagrasib) activates Hippo signaling, which in turn creates resistance against the inhibitors (PMID: 37729426). Since this article, it has been shown that CAF-secreted factors enhanced EMT signaling and promoted proliferation. This clearly suggests CAF might play a role in enhancing Hippo signaling, which also activates EMT in KRAS-driven cancers.  This aspect should be addressed as one of the future topics of this study to be followed up on. Add a few lines to the discussion on this topic by referring to the relevant references above. 

Authors: We agree with the reviewer suggested hypothesis and added a paragraph about the correlation between CAF-secreted factors, Hippo signaling and EMT in KRAS-driven cancers in the discussion, with the relevant references (pages 15-16, lines 516-533).

- Comments 2: CAF has also been known to regulate cancer metabolism (PMID: 33540679), and oncogenic KRAS mediates altered tumor metabolism. This aspect is another topic that might be followed up on in future studies generated from the current article. Authors should add a few lines on this by adding a few lines and the relevant mentioned reference.

Authors: Indeed, this is a relevant topic of discussion, particularly taking in consideration the downregulation of cholesterol homeostasis depicted in our RNAseq results, when comparing the samples treated with conditioned media of fibroblasts versus control treatment. We appreciate the reviewer suggestion and included a paragraph in the discussion regarding KRAS influence on tumor metabolism (page 16, lines 534-550).

- Comments 3: Figure 5, use different colors in the graphs to make the prominent distinctions between variables. 

Authors: We agree with the reviewer's suggestion. Not only did we add the color versions of the enrichment plots and tables, but we also applied a color code to the variables. We took the opportunity to use the same color scheme for the previous figures as well, improving interpretation. The new color code is as follows: siControl - grey; siControl treated with conditioned media from fibroblasts - dark grey; siKRAS - pink; siKRAS treated with conditioned media from fibroblasts - blue.

Reviewer 2 Report

Comments and Suggestions for Authors

1. In the title, please specify the influence and, if your research focused only on CRC, define the study area.
2. Move the simple summary section to either the introduction or conclusion. Also, improve the English in the simple summary.
3. In the abstract, the conclusion sentence lacks clear take-home information. Provide an analytical description of your findings instead of just listing them. The graphical abstract is very clear; please write an analytical description according to your excellent TOC.
4. I highly recommend making the introduction more engaging with a storytelling style. Avoid chopping the introduction into several sentences.
5. Please add subnumbers to subsections.
6. Move your tables to the results section.
7. Rewrite the conclusion.
8. Revise the results and discussion sections to be more narrative, guiding your readers through your findings.
9. Remove figure 6 if it is not mentioned in your manuscript.

Comments on the Quality of English Language

n/a

Author Response

Overall: We appreciate the reviewer's overall positive feedback regarding the quality of our work. We thank the input that was given and we have addressed the several points raised by this reviewer.

- Comments 1. In the title, please specify the influence and, if your research focused only on CRC, define the study area.

Authors: As suggested, we have changed the title and specified the study was done in CRC cell lines.
      Previous title: Unveiling the Mechanisms to Bypass KRAS Inhibition: In Vitro Insights into the Influence of Fibroblast-Secretome.

      New title: Fibroblasts promote resistance to KRAS silencing in colorectal cancer cells.

- Comments 2. Move the simple summary section to either the introduction or conclusion. Also, improve the English in the simple summary.

Authors: The simple summary should be a section on its own according to the journal guidelines, so we couldn’t follow the suggestion to move it to neither the introduction nor the conclusion. However, we did improve the English as requested, and rewrote this section to be more engaging (page 1, lines 18-28).

- Comments 3. In the abstract, the conclusion sentence lacks clear take-home information. Provide an analytical description of your findings instead of just listing them. The graphical abstract is very clear; please write an analytical description according to your excellent TOC.

Authors: We appreciate the praise regarding your graphical abstract. As suggested, we rewrote the abstract in a more analytical way to better convey the importance of the work (page 1, lines 29-44).

- Comments 4. I highly recommend making the introduction more engaging with a storytelling style. Avoid chopping the introduction into several sentences.

Authors: As per your suggestion, we rewrote the introduction chapter with a more a story telling style. We also added more content and relevant references to improve the quality of information given (page 2-3, lines 53-111).

- Comments 5. Please add subnumbers to subsections.

Authors: We appreciate this suggestion, since it helps guiding the reader through the text more easily. As you can confirm, the subsections from both the material and methods, as well as the results are now subnumbered.

- Comments 6. Move your tables to the results section.

Authors: We believe the reviewer is referring to the supplementary table (Table S1), as the only other tables we present are in the Materials and Methods section. The table in question refers to the RNAseq results and consist of an Excel file with more than 4000 entries. Incorporating such an extensive list of genes directly into the text would be impractical. We hope the reviewer agrees that presenting this large dataset as an Excel file is the most effective format for conveying these results.

- Comments 7. Rewrite the conclusion.

Authors: Following the reviewer’s suggestion, we have rewritten the conclusion to be more impactful and aligned with the reviewer's expectations (page 16, lines 563-572).

- Comments 8. Revise the results and discussion sections to be more narrative, guiding your readers through your findings.

Authors: We thank the reviewer for the suggestion and we thoroughly revised both the results and discussion. Besides improving the narration style, we also enriched the conclusion with further explanation points of our results (pages 7-16, lines 272-561).

- Comments 9. Remove figure 6 if it is not mentioned in your manuscript.

Authors: We thank the reviewer for pointing out this lapse. We have now mentioned figure 6 in the discussion section (page 16, line 553).

Round 2

Reviewer 1 Report

Comments and Suggestions for Authors

All concerns are addressed and the paper is ready for acceptance.